# Automatic Landing of Unmanned Aerial Vehicles via Wireless Positioning System with Pseudo-Conical Scanning

**DOI:** 10.3390/s22176451

**Published:** 2022-08-26

**Authors:** Ilia Iliev, Ivaylo Nachev

**Affiliations:** Faculty of Telecommunications, Technical University of Sofia, 1000 Sofia, Bulgaria

**Keywords:** unmanned aerial vehicles (UAV), automatic landing, pseudo conical scanning, radiolocation, navigation, phased antenna array

## Abstract

In this work, a wireless UAV unmanned landing system is considered, using the principles of pseudo conical scanning with a phased antenna array (PAA). The basic requirements for the characteristics and parameters of the system as a whole and of its components are defined. Special attention is paid to the primary sensor of the system—PAA with electronic scanning. A variant with a minimum number of four states on the radiation pattern of a low-budget patch PAA was studied. A linear regression of the difference characteristics of the measured radiation beams is proposed, which allows the practical application of the recommended landing algorithm with low computational complexity. Systematic and random positioning errors, both by measurement and by Monte Carlo simulation, were studied. Obtained statistical results prove the algorithm convergence and acceptable accuracy for the system implementation. They are applied if necessary to adjust the Kalman filter parameters. The proposed wireless system can be used for unmanned landing, tracking, and navigating the UAV in flight, or wireless navigation of other mobile objects.

## 1. Introduction

Unmanned aerial vehicles (UAVs) are a special apparatus that in recent years are increasingly used to solve problems in industry, telecommunications, agriculture, ecology, military affairs, civil defense, transport, entertainment, etc. [1,2]. It is essential to automate the UAV operating processes. The automated landing of UAVs expands the possibilities and scope of their use. Typically, qualified operators perform UAV landings. On the other hand, these devices have the necessary technical tools, optical and other onboard sensors, which are used for landing.

Most existing UAV automatic landing techniques are based on differential GPS systems [3], pseudoconical scanning radio systems (PCS) [4], laser optical curtains [5,6], video systems, image recognition at the landing site [7], etc.

Usually, the landing pad is the same size as the UAV. In some cases, the civilian GPS accuracy is often insufficient to perform a landing process. Additionally, the landing must be performed on mobile vehicles: a boat, another vehicle, or enclosed areas.

Different positioning methods have their advantages and disadvantages. For example, Real-Time Kinematic (RTK) [8] and/or Differential GPS systems have high precision. They require an additional base station for more accurate navigation and communication with moving objects. The additional equipment, in many cases, has a cost comparable to the UAV. In addition, RTK uses the connection to satellite navigation systems and a base station, which may, in some cases, be inapplicable. In patents [5,7], the landing is performed by analyzing an image recognition of a mobile landing pad. Optical laser methods also have the necessary accuracy, but they have a limited range of capture and tracking of the object and are not applicable in bad weather conditions: snowfall, fog, or landing in areas with dense smoke.

Landing methods with radio signal processing and pseudo-conical scanning (PCS) are known for the imposition of the radar and navigation basic principles. They are widely used in automatic satellite docking, for satellite tracking from terrestrial telecommunication stations and tracking and navigation of mobile objects [9,10].

The PCS’s main element is the system sensor—phased antenna array. The unmanned landing system’s accuracy, range, and functionality depend significantly on its parameters. The main advantages of PCS are:It is possible to track and navigate remote objects at relatively bigger distances compared to optical systems. Restrictive condition is the radio connection energy balance with the mobile object.Satisfactory accuracy for navigating, directing, and performing the automatic landing process. Due to the nature of PCS, it is interesting to note that positioning accuracy increases with decreasing distance between UAV and landing pad.The landing pad may be of dimensions commensurate with the UAV size.Ability to work in bad weather, rain, snow, smoky areas, etc. Of course, it depends on the selected operating frequency, which affects the propagation of electromagnetic waves.Possibility to land on mobile platforms.Relatively low cost and easy implementation.

The electromagnetic wave multi-path propagation has a significant effect on the accuracy even in the presence of obstacles. The received signal power can fluctuate in the range of 3–6 dB. Hence one of the method disadvantages related to the requirement for direct visibility between the moving object and the landing site. Then, the propagation model can be interpreted as a Rice fading model. This effect can be minimized if the PCS antenna array has a low level of the side radiation lobes. Moreover, as the distance between the UAV and landing site decreases, the positioning accuracy increases, and direct path signal strength increases compared to the reflected ones.

In this paper, a system solution is analyzed based on the UAV’s automatic steering to the center of a stationary landing site by a modified pseudo-conical scan version.

## 2. Materials and Methods

### 2.1. Pseudo-Conical Navigation System

A pseudo-conical navigation system is a separate autonomous radio communication system used to determine the relative position of an object in space. The UAV landing process is performed as follows. The UAV is directed to the landing site through the available tools and algorithms for navigation and positioning, usually using GPS. Once the UAV falls within a PCS system range, the automatic landing algorithm is started. The algorithm should include the following main processes: determining the UAV location relative to the landing pad (LP) coordinate system, tracking, filtering, and steering to the coordinate system center, as shown in Figure 1.

The automatic landing system is conditionally divided into transmitting and receiving parts. An interesting question is the location of the two parts. The following options can be formulated here:The transmitting part is located in LP. The receiving part is on a UAV, and it uses PCS. All tracking and positioning processing is performed in the UAV, and these functions can be integrated into the vehicle’s onboard algorithms. The disadvantage of this approach is that the receiving part antenna system has more weight than the transmitting part because it has to perform a pseudo-conical scan. Additionally, the receiving antenna dimensions and its location will affect the flight characteristics. The vehicle energy source must have a larger capacity to provide energy for the operation of the receiver and UAV in landing mode. This requirement also applies to other options. Energy must be provided for the transmitting or receiving part of the PCS system.The UAV has a transmitting part, and the receiving part is in the LP. This option is more suitable according to the above considerations, but it requires landing positioning commands to be transmitted from the LP via a communication channel. That is not a significant problem because remote control channels can be used and keep the same device positional data format. The positioning and landing algorithms can then be integrated into the remote control module.The radiated UAV radio signals can be used instead of having a separate unique transmitter. They usually bear system messages and/or video streams generated by a camera located on the device. Then, only the receiving part of the landing site is needed. This option significantly reduces the whole system complexity: no need to install additional UAV equipment, it does not increase the UAV weight, and it reduces the system cost.

PCS may be performed in the transmission part, but then it is necessary to transmit synchronization information between the two parts. The receiver must know which antenna beam has transmitted a signal to determine a position. This condition only complicates the system and is not acceptable for automatic landing.

Below is described option two of the system with PCS in the receiving part, including option three—Figure 2.

The positioning system works as follows. After the device enters the automatic landing system operation area, the UAV radio transmitter turns on. It forms a reference radio signal with a specific power. The signal is radiated from the transmitting antenna and is used to determine the location on the receiving side. Additionally, through the appropriate modulation and coding, it can carry navigation information, identification information, and more. The receiving part is in the landing site. The system sensor is a phased antenna array, and its diagram successively in time scans the space in a circle with a discrete finite number of positions in space—Figure 1. The diagram describes a cone with a certain central angle 2***Θ_0_***. The receiving power is measured for each beam state, and after processing and calculations, the angular coordinates of the object in space are determined. These coordinates give the device deviation from the LP center and are used to form commands to correct the UAV location. A unique algorithm performs the automatic landing process with the estimated angular deviations. The UAV location is determined by the angular coordinates and its height above the ground. This information is taken from the UAV navigation messages. Landing commands are transmitted on the available UAV communication channels, and with each iteration, the device consistently reduces its height in space.

#### 2.1.1. Transmitting Part

The transmitting part consists of a radio transmitter RTr and an antenna *A_tr_*. For positioning, the transmitter forms a reference radio signal with a certain power, and the signal can carry additional information for navigation, identification, synchronization, etc. The transmitting signal power must be such as to ensure the range and accuracy of the landing system.

The system operating frequency is an important parameter. It determines the operation distance and angular range, the transmitting and receiving antenna feasibility, the ability, and accuracy for the angular coordinate assessment, equipment weight, implementation, cost, etc. In terms of scanning functionality and sensor dimensions, it needs to be in the range of dm, cm, or mm waves. This frequency can be selected in the free licensed frequency bands (ISM). There are several considerations here:The antennas size, weight, and radiation patterns feasibility largely depend on the operating frequency.The electromagnetic wave propagation attenuation depends on the wavelength. That significantly affects the radio link budget. Free licensed frequencies also are limited by the maximum isotropic radiated power of transmitters. That limits the operating range of the positioning system.The UAV radio signals emitted for its operation can be used. For example, navigation and control signals, signals related to the transmission of video streams, etc. This feature should be used because it has many advantages. No additional PCS transmitter is required to be mounted on the UAV. Apart from the cost, this is especially important for maintaining the aircraft’s flight characteristics, related to payload weight, energy consumption, flight distance, etc.

Furthermore, in the presented discussions, the ISM band of 2.4 GHz is chosen. The choice was made for the following reasons. Easy feasibility of the antenna array with PCS, low cost, achieving the required system operation range, instead of an additional transmitter can be used the UAV transmitter for information exchange with the remote controller RC.

The transmitting antenna Atr is mounted on a UAV and must be small in size and weight. The Atr characteristics must meet the following conditions. The radiation diagram has a specific width and shape. The width defines the PCS system angular range. If it operates in an angular range ±***Θ_0_*** (Figure 1), then the diagram width must not be less than 2***Θ_0_***. This condition is necessary because the electromagnetic wave radiated power must not change when changing the UAV angular location. The received signal power will not depend on the device’s angular location when the height is constant. For ease of implementation and low weight and dimensions, the polarization of *Atr* is chosen to be linear. Dipole, printed, patch antennas with cosine RP are suitable. A dipole with a gain factor is chosen for the initial design calculations *G_tr_* = 3 dBi. The choice is justified because most UAVs have a built-in dipole antenna in the discussed frequency. Patch antennas have a higher gain in the range of 6–8 dBi and are suitable for increasing the system range.

The output transmitted reference signal power shall comply with the limit for the maximum permissible EIRP set by the regulatory authorities in the respective country for the ISM band use. According to the selected factor *G_tr_*, the transmitter power is chosen to be *P_T_* = 10 dBm.

#### 2.1.2. Receiving Part

The receiving part must perform functionalities for determining the device location, tracking, filtering, and steering to the coordinate system center. The necessary elements are an *A_re_* antenna array with pseudoconical electronic scanning, RRe radio receiver, BmC block for control of the radiation diagrams, EAD power meter, EST block for determining the angular coordinates *ϕ_S_*, *θ_S_*, PC positioning controller, and RC remote control controller through which navigation information is exchanged with the UAV.

The phased antenna array (PAA) with pseudo-conical electronic scanning *A_re_* is an essential element referred to as the system sensor. Using control signals from the BmC, the antenna forms a finite number of radiation patterns (beams). Each beam deviates at a certain angle from the coordinate system *z*-axis ***Θ_0_***—Figure 1. The BmC block sequentially switches the antenna diagrams.

The RRe radio receiver’s task is to process radio frequency signals, providing the necessary sensitivity and selectivity by amplification, filtering, and, if necessary, frequency conversion and/or detection. A radio receiver with analog processing of radio frequency signals, built on a superheterodyne principle or with direct frequency conversion, can be used here. An option is to use a software-defined radio. If it is not necessary to detect the information borne by the reference radio frequency signal. In that case, the receiver may contain a radio frequency bandpass filter with a high attenuation for frequencies outside of the bandpass, a radio frequency amplifier providing a sufficient level for operation of the power measuring unit. This option has limited sensitivity and low dynamic range but is acceptable due to its simplicity and low cost.

The EAD unit is part of the radio receiver and is an envelope detector, at the output of which there is a signal proportional to received power. In SDR, the power estimation is calculated naturally after the respective processing of in-phase and quadrature components. This block can be performed with a logarithmic power meter for the simplified version. It is an analog integrated circuit whose output has a voltage proportional to the received power on a logarithmic scale.

In EST block the signal, which is proportional to the received power and after analog-digital conversion, is used to determine the object angular coordinates *θ_S_*, *ϕ_S_*. This unit is a hardware microcontroller in which the received power of the separate antenna beams is processed using a corresponding algorithm, and from this, the angle coordinates are determined. This can be done by prior knowledge of the antenna radiation diagrams, which give the power dependence on angular coordinates.

The UAV location is relative to the Cartesian coordinate system (O*x,y,z*), where zero is in the landing pad center, the antenna sensor center. The polar coordinate system (O*ϕ*θz) is used for convenience in calculations. The connection of two coordinate systems is determined by angular object coordinates being *ϕ*—the angular deviation in the xOz plane and *θ* in the xOz plane. The position of the UAV in space is uniquely defined by the coordinates of a point *M (ϕ,θ,z)*. The transformation between two coordinate systems is:(1)$$O\phi \Theta z\;\underset{}{\to }\;Oxyz\;x=z.tg\left(\phi \right),\;\;y=z.tan\left(\Theta \right),\;z=z,\;Oxyz\underset{}{\to }\;\;O\phi \Theta z\;\phi =atan\left(x/z\right),\;\;\Theta =atan\left(y/z\right),\;z=z.$$

The block EST *ϕ_S_*, *θ_S_* calculates a projection of the point M in the plane O*ϕθ*. The UAV position is determined by the estimated angular deviations *ϕ_S_*, θ_S_, and current height z—Figure 3.

The positioning controller (PC) is part of a microprocessor. Its software calculates the corrections for steering the device to the coordinate system center and implements the landing algorithm with estimated angular deviations *ϕ_S_*, *θ_S_*, and current height *z_S_*. The landing algorithm tracks the location and performs necessary data processing by Kalman filtering [11,12]. The landing corrections are transmitted as commands to the UAV via R—remote control.

#### 2.1.3. Phased Antenna Array with Pseudo-Conical Electronic Scanning

PAA with PCS is the main system sensor with which the unmanned landing is realized. The sensor basic technical requirements are defined as follows:Operating frequency. The operating frequency is determined according to the given system frequency considerations.PAA with circular polarization. Thus, the UAV can use a transmitting antenna with linear polarization. That minimizes the received power dependence on the difference in the UAV directions coordinate systems and landing pad.The number of beams for pseudo-conical scanning. They determine the positioning accuracy and the algorithm complexity for estimating angular deviations and the sensor complexity and cost. With many beam states, an increase in the angular coordinates estimating accuracy is expected. However, on the other hand, this leads to antenna sensor complexity, such as an increased number of elementary antennas in PAA, increased number of additional elements for their excitation, complex implementation, larger antenna dimensions, and increased cost. Therefore, a compromise is sought to meet the conflicting requirements outlined above. The number of beams of the antenna array is chosen to be at least equal to 5, with one central beam and four deviated in the coordinate system directions—Table 1.
sensors-22-06451-t001_Table 1Table 1States and deviations of the beams.Antenna Beam State *i*Angular Deviation of the Beam in Both Planes [deg]0, Central (Null) position *ϕ_o_* = 0°, *θ_o_* = 0°1, Back direction (B)*ϕ_o_* = 0°, *θ_o_* = −20°2, Right direction (R)*ϕ_o_* = 20°, *θ_o_* = 0°3, Forward direction (F)*ϕ_o_* = 0°, *θ_o_* = 20°4, Left direction (L) *ϕ_o_* = −20°, *θ_o_* = 0°Beam deviation angle ***Θ_0_***. The angle determines not only the PAA complexity but also the UAV angular engagement range before positioning. This parameter also determines the system’s minimum range of acquisition—the distance from which the PCS tracks and UAV positions. In Table 1, we can see that the PCS angular deviation is *Θ_0_* = 20°, and the system angular range is 2***Θ_0_*** = 40°. Assuming that the UAV is in flight mode and is guided to the landing site by some navigation system with coordinate uncertainty ∇x,y, then the minimum system engagement distance is determined by the formula:(2)h0min=∇x,y/tgΘo,
For example, if the UAV positioning system uses GPS with a relatively large value of ∇x,y=6 m for the altitude, h0min = 16.5 m.The radiation pattern width of each beam depends on the diagrams of each PAA element, its number, and position. All this also affects the antenna gain. The beams should be symmetrical in two antenna planes *xOz*, *yOz* to simplify the signal processing. That means the number of elementary radiators must be the same in both *Ox*, and *Oy*, for example, 2 × 2, 4 × 4, etc. On the other hand, a narrow beam diagram implies a high characteristic steepness and hence more high accuracy. There is the problem of sensitivity in space areas where the PAA does not have sufficient gain. A compromise of Δ*ϕ_max_* = 20°, Δ*θ_max_* = 20° for all beams is chosen. Research shows that then the diagram in the falling segment has a steepness of 2 dB/grad, which is sufficient for small distance positioning.Satisfactory amplification to achieve system sensitivity. A compromise shall be chosen between sensitivity and accuracy, and the PAA gain shall be not less than 6 dBi.Low level of the side lobles of each antenna beam states. Recommended to be no bigger than −10 dB.Compact dimensions.High-speed antenna beam switching.High reliability.Cheap realization.

According to the selected operating frequency, the above requirements can be achieved if the PAA single radiation element is a patch antenna. Advantages of patch antennas are simple design and implementation methodology, low cost, high gain, easy to achieve circular polarization, and ease of integration into an antenna array. The PAA contains patch emitters and an excitation network in a microstrip implementation [13,14]. The antenna array and excitation network can be implemented as a multilayer circuit board, which in turn provides the compact size of the PAA.

To prove the relevance of the proposed approach and system for UAV positioning and landing, a PAA with 2 × 2 deployed circular patch elements was designed, simulated, and measured, as seen in Figure 4.

The antenna is designed and fabricated with dielectric substrate FR4, with relative dielectric permittivity $\varepsilon_{r}$ = 4.75, tan δ = 0.003, and height h = 1.5 mm for the operating frequency $f_{c}$ = 2.4 GHz. The wavelength in free space is λ = 125 mm and distance between the element centers is obtained at 0.5λ [15,16,17]. The circular polarization is done by two feed points for each patch located at 90° electrical degrees concerning each other and a suitable dephasing obtained from the feeding network. The realized antenna has nine beam states, the first five are according to Table 1. The switching of the different PAA diagrams is performed with electronic switches and microstrip lines in the excitation network structure.

The measured results for the normalized antenna gain after PAA design and fabrication are given in the following figures. The automatic system [18] performing antenna RP measurement results. Figure 5 shows the measured four states from 2 to 5 in a 3D coordinate system. The normalized gain diagrams, measured in logarithmic scale in the Cartesian coordinate system, are shown in Figure 6a,b for the two antipodal states concerning *θ*, and Figure 7a,b for the states concerning *ϕ*, respectively.

#### 2.1.4. UAV Angular Coordinates Determination

In the EST algorithm’s output, the object’s angular coordinates—*ϕ_S_*, *θ_S_*—should be calculated. The measured values of the received power at different PAA beam states must be used. There are various methods for estimating angular coordinates from measured powers, such as with Least Mean Square Error (LMSE), the correlation between measured power values and the angular coordinates of the diagram, Kalman filtering, etc. For example, in [16], the two LMS and Kalman filtering methods are discussed. In [17], a least-square error (LSE) estimation method is used. The main difference with the present work is that the cited systems for moving objects tracking are determined by their application specifics:They are intended for tracking satellites that are at a vast distance from the ground station—hundreds, thousands of kilometers, and in the considered system, the distance between the object and the sensor is dozens of meters.They use a conical scanning antenna with a narrow RP of a few degrees and have high amplification to achieve the required signal-to-noise ratio. The scanning antenna pattern is deflected to a small angle due to the long distances, and the order of deviation is a mdeg. For the used sensor, the beam deviation is dozens of degrees because considerations given in Section 2.1.3.Conical scanning is accomplished by the mechanical movement of a parabolic antenna. The number of beam states and measured powers is large for higher positioning accuracy. Here, the number of beam PAA states is proposed to be limited to four in the four directions in the antenna plane due to the imposed requirement of the PAA simplicity and the slight computational algorithms complexity.The narrow diagram parabolic antenna allows applicating an analytical beam model approximated by a Gaussian function. It estimates the object angular deviations by giving a direct mathematical, deterministic analytical relationship between measured powers and satellite angular coordinates to the antenna axis. For the considered PAA with electronic scanning and a small number of elementary emitters, the diagram for the corresponding beam is inaccurately approximated by a Gaussian function. Additionally, the side lobes level is higher, and such an approximation will increase the estimation error.

Let us suppose that the antenna gain as an angular coordinates function is approximated by a Gaussian function [16]. This approximation is relevant, provided there are no side lobes on the radiation pattern. The normalized gain $g_{i}\left(\phi ,\;\Theta \right)\;$ at the *i-th* beam state (*i-th* radiation pattern) is described by:(3)$$g_{i}\left(\phi ,\;\Theta \right)=exp\left\{-4\;\ln\;2\times \left[\frac{{\left(\phi -\phi_{0i}\right)}^{2}}{\Delta \phi_{i}{}^{2}}+\frac{{\left(\Theta -\Theta_{0i}\right)}^{2}}{\Delta \Theta_{i}{}^{2}}\right]\right\}$$
where $\phi_{0i}$ and $\Theta_{0i}$ are the projection of the maximum of the diagram in the 2D angular coordinate system, and $\Delta \phi_{i}{}^{}$ and $\Delta \Theta_{i}$ are the beamwidths at a gain of 0.5. The antenna is designed such that for each beam *i* $\Delta \phi_{i}{}^{}=\Delta \Theta_{i}=\gamma$.

If the logarithmic scale of $g_{i}\left(\phi ,\;\Theta \right)$ is used, the gain is:(4)$$F_{i}\left(\phi ,\;\Theta \right)=-4log_{10}2\times \left[\frac{{\left(\phi -\phi_{0i}\right)}^{2}}{\gamma^{2}}+\frac{{\left(\Theta -\Theta_{0i}\right)}^{2}}{\gamma^{2}}\right]\;[\mathrm{dB}].$$

The simplification of Equation (4) can be achieved if, for some *i*, the beam center projection in the angular coordinate system lies on its axes. For scanning, let only these states be used. For the projected PAA and *i* = 1,2,3,4 according to Table 1, difference diagrams in pairs will lead to a linear dependence of the angular coordinates or:(5)$$F_{2}\left(\phi ,\;\Theta \right)-F_{4}\left(\phi ,\;\Theta \right)=4log_{10}2\times \frac{\phi_{02}}{\gamma^{2}}.\phi \;[\mathrm{dB}],\;F_{3}\left(\phi ,\;\Theta \right)-F_{1}\left(\phi ,\;\Theta \right)=4log_{10}2\times \frac{\Theta_{03}}{\gamma^{2}}.\Theta \;[\mathrm{dB}].$$

Equation (5) describes two planes on which the projections of their normal vectors onto the 2D angular coordinate system are orthogonal, and the normal vectors themselves are orthogonal. Furthermore, the two planes’ offset, concerning the coordinate system center, is zero. They show that object angular coordinates can be easily calculated based on the measured powers on a logarithmic scale For this purpose, (5) can be transformed into:(6)$$P_{2}\left(\phi ,\;\Theta \right)-P_{4}\left(\phi ,\;\Theta \right)=F_{2}\left(\phi ,\;\Theta \right)-F_{4}\left(\phi ,\;\Theta \right)\;[\mathrm{dB}],\;P_{3}\left(\phi ,\;\Theta \right)-P_{1}\left(\phi ,\;\Theta \right)=F_{3}\left(\phi ,\;\Theta \right)-F_{1}\left(\phi ,\;\Theta \right)\;[\mathrm{dB}],\;\mathrm{because}\;\mathrm{for}\;\mathrm{every}\;i\;$$
(7)$$P_{i}\left(\phi ,\;\Theta \right)=P_{0}+F_{i}\left(\phi ,\;\Theta \right)\;[\mathrm{dBm}].$$

The power entering into the antenna aperture $P_{0}$ [dBm] is the same for all beam states. 

#### 2.1.5. Approximation of Measured Sensor Diagrams

The EST algorithm must have low computational complexity to be implemented as program code in a microcontroller. One option is to use measured beam antenna gains to find the angular coordinates’ inverse dependence on measured powers. The Gaussian approximation of diagrams and (5), (6), and (7) shows that such an approach is possible. It should be noted that the sensor would then have a systematic error due to Gaussian approximation error—diagrams are not Gaussian functions, a presence of side lobes, the designed antenna conditions in Table 1 are not strictly satisfied, the widths of the diagrams may be different in two planes of ϕ and Θ, also for the individual beams, or quadrature imbalance, since the approximated projections of the normal planes such as (5) are not orthogonal.

The measured PAA gain diagrams for the four states are shown more fully in Figure 6 and Figure 7. The difference diagrams in pairs are easily obtained from them. They are presented in Figure 8a,b.

The *F_R_*-*F_L_* difference diagram accounts for the variation of *ϕ*, and *F_R_*-*F_L_* for *θ*, and their indices are according to Table 1. The figures clearly show that an approximating plane in the sensor range ***Θ_0_*** = ±20° can be used, instead of the complex surfaces of the difference diagrams. The approximating planes are easy to describe mathematically, and their equations can be used to calculate the angular coordinates quickly. This idea is further developed below.

Because numerical values of the diagrams are known, the plane equations can be determined by regression analysis. The plots show that the spacing characteristics are linear functions on the angular coordinates only in regions around the sensor center. Furthermore, the sensor captures and tracks the moving object in the angular range of ***Θ_0_*** = ±20°. Then, the difference diagrams regression is limited to the specified angular range. Figure 9 shows the results of one such linear regression.

In addition, Figure 8 gives the regression planes plotted on the measured differences diagrams.

The regression was performed using MATLAB (Matlab R2022a/9 March 2022, The MathWorks, Inc. [9]—3 Apple Hill Drive, Natick, MA 01760 -508-647-7000) [19,20]. A first-degree polynomial surface fit on ϕ and Θ with the regression method Least Absolute Residuals (LAR) was used. The standard errors of the regression for Figure 9a,b are rmse(*ϕ*) = 0.24 and rmse(θ) = 0.23 degrees, respectively.

The equations of the approximated planes are represented as follows:(8)$${\left|\begin{array}{c} Ap_{\phi }=a.\phi +b.\Theta +o_{\phi }\\ Ap_{\Theta }=c.\phi +d.\Theta +o_{\Theta } \end{array}\right.}_{}.$$

As a result of the approximation, the following coefficients were obtained: a = 0.9002; b = 0.1007; c = 0.0403; d = 0.9351. Because the antenna is centered concerning its coordinate origin, offsets $o_{\Theta }=o_{\phi }=0$. The centering is done in the process of antenna measuring. Because the coefficients b and c are different from zero, a quadrature imbalance exists. The diagrams of the fabricated antenna deviate from the requirements in Table 1. The angular coordinates cannot be directly calculated with (5). It is appropriate to solve the above system of equations for the angular coordinates by inverse matrix or consistently with: (9)$$\Theta_{s}=(Ap_{\Theta }-\frac{c}{a}Ap_{\phi })/D,\;\;\;\;\;\;\;\phi_{S}=(Ap_{\phi }-b\Theta_{s}{}_{})/a,\;\;\;\;\;\;D=d-b\frac{c}{a}$$

Figure 10 presents the differences functions for the two cross-sections along the *ϕ* and *θ* axes and the approximating lines from the resulting cross-sections. 

The figure shows once more that the interval ±20° they can be approximated by linear functions.

#### 2.1.6. Positioning and Landing Algorithm

The main algorithm requirement is low computational complexity to be implemented as program code in a simple microcontroller.

The algorithm uses the estimated angular coordinates to correct the UAV position to the coordinate system center—LP center. The angular coordinates are computed from the measured power values of the received scan signal with the different antenna array diagrams. A vector consisting of the measured powers is input to the internal EST algorithm:(10)$$P=\left[P_{1},P_{2},\ldots ,\;P_{N}\right],$$
where *N* is the beam numbers.

From the EST, the object angular coordinates—*ϕ_S_*, *θ_S_*—estimated by the receiver part are obtained. The vehicle’s location is determined by the estimated angular deviations *ϕ_S_*, *θ_S_*, and the vehicle current altitude *z = h*.

The UAV is directed to the landing site through available tools and algorithms for navigation and positioning. Once the UAV is within range of the system with the PCS, an automatic landing algorithm is started. As defined, for the UAV to fall within the sensor area, the minimum high must not be less $h_{0min}$. Each position *k* of the UAV is defined by the coordinates $\phi_{k},\Theta_{k}$, *z_k_ = h_k_*. The current high *h_k_* is an input parameter and is taken at each step by the vehicle’s onboard navigation system. At each lending process step, the algorithm adjusts a vehicle position in the plane (x0y) and decreases the absolute high by a step *z_k+1_ = h_k_ +*Δ*h*. The change in height determines the landing speed and is chosen experimentally. Acceptable values of Δ*h =* 0.5 m to 2 m depending on the current high *h_k_* and UAV flight characteristics.

The algorithm must perform the following basic steps:Evaluating the current height *h_k_* and step counter *k* = 0.Sequentially switching the sensor diagrams for each *k* and measuring the received power.With the measured powers $P_{k}=\left[P_{0k},P_{1k},P_{2k},P_{3k},\;P_{4k}\right]$, the angular coordinates of UAV $\phi_{Sk},\Theta_{sk}$ are calculated on the basis of (6, 9). Here $P_{0k}$ is the zero state of the antenna array diagram, and respectively $P_{3k},P_{5k}$ are the measured powers in diagrams with deviations +20°, −20° along the x-axis, and respectively, $P_{2k},P_{4k}$, +20°, −20° along the y-axis;Calculating the Cartesian coordinates with Equation (1).Kalman filtration and discrete PI controller. It is applied to minimize the mean square error of the estimation of the angular coordinates $\phi_{k},\Theta_{k}$. When setting filter parameters, the sensor error estimate shown in the next section is used. The deviation of the UAV from the coordinate system coincides with the error function of the PI controller, and the position correction command is formed by:(11)$$x_{k+1}=K_{I}\times x_{k}+K_{P}\times z_{k}\times tg\;\left(\phi_{Sk}\right),\;\;\;\;y_{k+1}=K_{I}\times y_{k}+K_{P}\times z_{k}\times tg\;\left(\Theta_{sk}\right),$$
where $K_{I},K_{P}\;$ are the integral and proportional coefficients of the PI controller. Here the height $z_{k}$ is not subject to filtration because it relies on the built-in processing in the UAV control system.Sending command to the UAV to change position in-plane (x0y) with the calculated by: $K_{P}\times z_{k}\times tg\;\left(\phi_{Sk}\right)$ и $K_{P}\times z_{k}\times tg\;\left(\Theta_{sk}\right)$;Decreasing the height *h_k_ = h_k_ +* Δ*h* and check if *h_k_* < ***hmin***. ***hmin*** is the minimum height to which the vehicle changes its vertical position during landing. It is selected depending on the characteristics of the UAV but cannot be less than the limit for the antenna’s far-field. For example, for the antennas selected above, ***hmin*** ≥ *1 m*.If not, a command to the vehicle to change *h* is transmitted.
Increasing the counter *k = k + 1*. A pause for a specific time, the UAV needs to be set to the new position. This time depends on the UAV flight characteristics and is consistent with the speed of correction and landing.Iterative execution from step 2 onwards.If yes, end the algorithm a landing command is transmitted.

## 3. Results

### 3.1. Assessment of the Approximation and Positioning Error

The use of approximating planes on the difference diagrams to estimate the angular coordinates is an approach that implies low computational complexity for the estimation algorithm. However, the measured antenna diagrams and their spacing characteristics are not linear over the entire angular range of the PAA. Therefore, it is necessary to estimate the systematic error from positioning.

The absolute error is calculated as the difference between the actual UAV angular coordinates *ϕ*, *θ*, and the coordinates *ϕ_s_* and *θ_s_* determined by the approximated characteristics.

Figure 11 separately presents the absolute error in degrees in the estimation of the two angular coordinates *ϕ_S_* and *θ_S_*. The same results are shown as family of characteristics for the two cases in Figure 12.

Table 2 gives specific values and intervals of the absolute error variation. The results show that the error for both coordinates is no larger than ±2° when the positioned object offset concerning the sensor center is ±15°. At the sensor center, it is no larger than 0.15°, which gives a bias of a few millimeters at an apparatus height above the landing site of *h* = 1 m. The error decreases as the object gets closer to the sensor’s center. This fact is important because it shows the approach’s applicability and will ensure convergence of the landing algorithm.

Separately, the above fact is confirmed by representing the vector magnitude with the estimated angular coordinates as a function of actual coordinates *ϕ*, *θ*—Figure 13.

Figure 13 demonstrates the positioning error by comparing actual angular distance circles and the curves determined using the proposed regression.

Figure 14 gives an estimate of the mean angular distance error and its standard deviation as a function of the actual angular distance *R*. The averaging was performed for *R_S_* estimate values over the circle defined by *R*.

Here it is once again proved that as the angular distance at which the navigated object is located decreases, the sensor accuracy estimate increases. The mean absolute error and its standard deviation in the range of angular deviation up to 20° are acceptable.

The estimated errors can be further used to determine the Kalman filter parameters.

### 3.2. Statistical Evaluation of the Impact of Power Measurement Noise and Coordinate Estimation

To test the proposed approach of automatic UAV landing using a wireless positioning system with pseudo conical scanning, a variant of a Figure 15 radio receiver was designed and fabricated. It contains a RF band-pass filter BPF, a radio frequency amplifier A, a power measurement block EAD, and a microcontroller. As mentioned, this variant has limited sensitivity and a small dynamic range but is acceptable due to its simplicity.

The BPF is the basic unit of the receiver, ensuring its selectivity and suppression of out-of-band interferences. A dual-coaxial-resonator filter structure with electrical coupling between the resonators is designed. Such a structure provides attenuation of not less than −30 dB for frequencies outside the passband at 100 MHz from the cutoff frequencies. The center frequency is the PAA operating frequency $f_{c}$ = 2.4 GHz, and the passband is BW ≈ 150 MHz.

The amplifier is needed to increase receiver sensitivity and output signal power to be within the dynamic range of the envelope amplitude detector (EAD). A ready-made low noise amplifier module was used, with a gain factor of 15 dB in the frequency range from 1 to 6 GHz.

The EAD block is an envelope amplitude detector implemented with a logarithmic power detector. An AD8318 RF Logarithmic Detector power meter (by Analog devices, Wilmington, USA) was used. A voltage proportional to the power of the input signal in relative value dBm is obtained in its output. This feature is important to reduce the algorithm computation to estimate the angular coordinates. The logarithmic detector dynamic range is from −65 dBm to +10 dBm. It matches the PAA’s received signal power and provides sufficient output voltage amplitude.

The resulting voltage is converted to discrete values by an analog-to-digital converter ADC. The same is embedded in the microcontroller. The micro-controller controls beam states through digital outputs and switches PAA beams for all cases. With each beam’s measured, discretized power values, the EST algorithm computes object angular coordinates *ϕ_S_*, *θ_S_*. The PC positioning algorithm produces the necessary commands to change the UAV location to the coordinate system center. The controller is implemented with an STN32F103C, which has a built-in 12 bits ADC and has sufficient speed to run the PAA beam switching, power measurement, angular coordinate estimation, and UAV positioning commands.

The transmitting part is mounted on the UAV and transmits an RF signal modulated with BPSK and direct spread spectrum (DSS). A transmit patch antenna with 6 dBi gain and a cosine diagram is used.

One measured value is obtained after averaging 2048 ADC samples. The rate at which the measurement is performed in nine states of the diagram per 100 ms. The power measurement for all PAA states, the angular coordinates estimation, and the control commands’ elaboration take a computational time tc no greater than 150 ms. 

The PAA *Are*, implemented and its parameters described in Section 2.1.3, is used as the system sensor.

The measurements were performed as follows. The transmitting part is mounted on a stand at different distances from the receiving part to be located in the center of the coordinate system, i.e., *ϕ = θ = 0*. The receiving part measures the received power and estimates the angular coordinates *ϕ*_S_, *θ*_S_. The received power was measured with five PAA diagrams (Table 1) at distances from 6 to 40 m between transmitting and receiving parts. This includes the variants where the UAV is located at ±20°.

After statistical processing, the random error influence of the power measurement can be determined, and the results are shown in Figure 16a,b. Figure 16a combines the measured powers *p* for different distances with different states of the PAA plot against the mean value Δ*p* = *p*-mean(*p*). The same figure shows the standard deviation as a function of the measured mean power (distance) and prediction interval with a 95% probability of falling a new measured power value. Figure 16b draws attention to the standard deviation dependence on the mean power.

The following conclusions can be made here:The standard deviation increases with decreasing received power, respectively, with increasing distance. Up to −45 dBm, it does not change and is of the order of 0.11 dB. The error is primarily determined at a small distance by quantization noise, switching noise, white thermal noise, and receiver noise figure.At longer distances, multipath propagation and Rice fading are manifested, and therefore an increase in the standard deviation is observed. The reason for this is also that the PAA diagrams have side lobes.The prediction interval for a 95% probability of hitting each new measured power value is related to the absolute error. It is ±0.21 dB for a small distance. The power difference characteristic deviation for the heaviest case gives ±0.37 dB. The value transformed by (9) into angular deviation is ±0.41°. For a distance of 6 m, the object coordinates deviation is 4 cm, and for 1 m, the deviation is 7 mm. At 40 m, the interval is ±0.86 dB, giving a 0.44 m offset of the coordinates from the centrum.The obtained standard deviation results can be used for the Kalman filter parameters tuning.

### 3.3. The Landing Algorithm Simulation Results

The landing algorithm was simulated in a MATLAB environment without using Kalman filtering. The integral and proportional coefficients of the discrete PI controller are chosen $K_{I}=1,K_{P}=0.55$ to ensure the system’s robustness. The initial height for a group of simulations is *ho* = 15 m and *ho* = 45 m, the step is Δ*h* = 0.5 m or the UAV has a certain number of positions in 3D space. The initial deviation in the *xOy* plane is different, but in this case, for *ho* = 15 m the deviation is ±25°, more than ***Θ_0_***, and for *ho* = 45 m ± 20°. The actual location is determined with the measured PAA difference characteristics, and the location correction is done with the approximated ones.

The influence of the error from the received power measurement is simulated by adding Gaussian noise with parameters $ℵ\left(0,std^{2}\right)$ to the measured power. The standard deviation of the power is taken from the results in Figure 16b and using the communication channel system parameters std = 0.14 dB for height 15 m and std = 0.45 dB for *ho* = 45 m. The study was performed using the Monte Carlo method.

The results given in Figure 17 refer to an initial altitude of *ho* = 15 m. Figure 17a illustrates the landing process relative to the 3D coordinate system of the landing site. Figure 17b shows the statistics of the coordinates of the estimated position a the end of the algorithm—*hmin* = 1 m above the sensor. The standard deviation is 0.7 cm, or the positioning accuracy in the final landing stage is satisfactory.

Figure 18 presents the results at a starting landing height of *ho* = 45 m. The worst case is investigated where the standard deviation of the measured power, according to Figure 16b, is std = 0.45 dB. The aggravation here comes from the choosing of Gaussian noise parameters. Its variance is not changed by the distance between the object and the sensor. Figure 18a shows the variation of the *x*, *y* coordinates in the landing process for the height *h*. Figure 18b gives a visual evaluation of the statistics at the end of the algorithm, as in Figure 17b for 1000 trials. From the *x* and *y* traces, the effect of measurement noise is visible. The standard deviation of the coordinates for the last point of the landing is on the order of std = 1.26 cm. A summary of the data is presented in Table 3.

Figure 17 and Figure 18 prove the applicability of the studied wireless landing system, even without Kalman filtration. The algorithm is converging, despite the presence of uncertainty in power measurement, and the system error of the positioning sensor is automatically involved in the tests performed.

**Table 3 sensors-22-06451-t003:** Study results of the standard deviation of the x, y coordinates at the end of the landing algorithm.

Start High *ho* [m]	Power Standard Deviation [dB]	x,y Coordinates Standard Deviation by *hmin* = 1 m
15	0.14	0.72 cm
45	0.45	1.26 cm

## 4. Discussion and Conclusions

The research shows that it is possible to implement a wireless positioning system for UAV unmanned landing using a PAA sensor with electronic scanning and a limited number of scanning beam states. It is necessary to observe the recommendations made to the characteristics and parameters of the system when selecting and designing its components. Pre-measurement of the radiation patterns and subsequent linear regression allows for achieving satisfactory accuracy even with a simplified version of the receiver part. The landing algorithm is convergent due to the systematic positioning error reduction as the landing site centrum is approached and the random error reduction from received power measurement as the UAV altitude is decreased. The error standard deviation increases with increasing distance due to electromagnetic wave multipath propagation and the presence of side lobes of the used PAA sensor. Therefore, a line of sight between the transmitting and receiving parts of the system is required during the landing process. Another constraint is that the algorithm must be run when the UAV is within the sensor angular range.

Tests of the algorithm have been carried out in simulation without applying Kalman filtering with artificial addition of uncertainty with a centered Gaussian distribution. The obtained statistical parameters allow the tuning of the Kalman filter characteristics. Its application will guarantee the algorithm’s convergence and reduce the positioning error at the final point of the landing process.

The developed system allows not only the unmanned landing of UAVs, but it can also find applications in flight mode: tracking of UAVs or wireless positioning and navigation of moving objects in 3D and 2D space.

An actual UAV landing study is to be carried out using the developed system and implementing Kalman filtering in the processing. Another problem to work on is to extend the angular range of the system by exploiting the central position of the PAA diagram and to reducing the positioning error as the number of PAA beams increases.

## Figures and Tables

**Figure 1 sensors-22-06451-f001:**
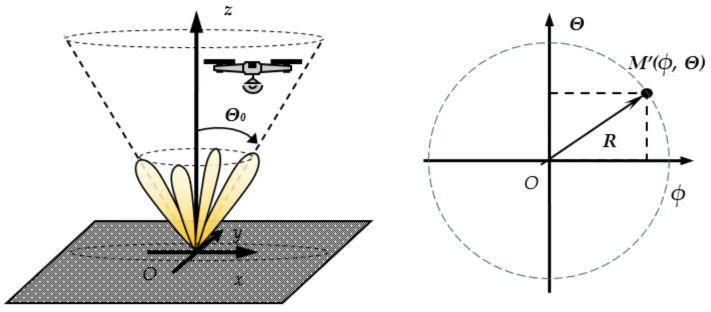
Coordinate systems of the pseudo-conical navigation system.

**Figure 2 sensors-22-06451-f002:**
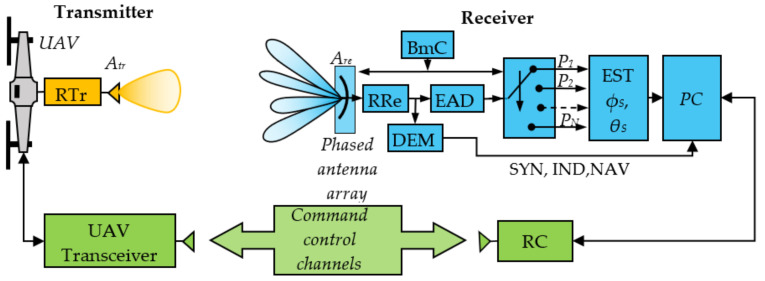
The system block diagram. Transmitting and receiving parts.

**Figure 3 sensors-22-06451-f003:**
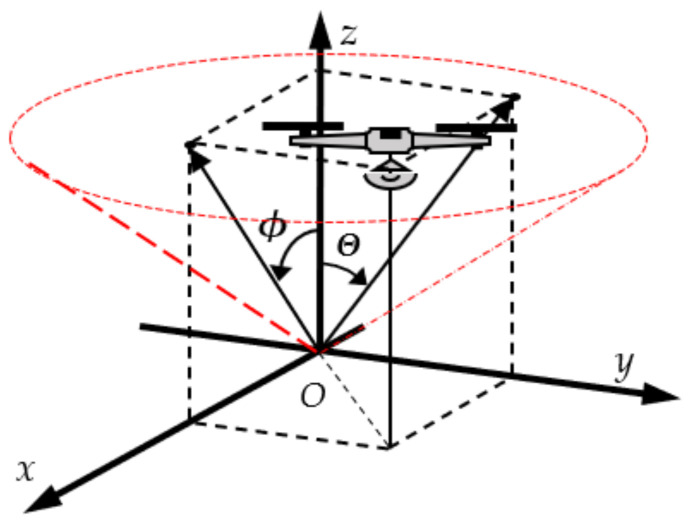
Polar and Cartesian coordinate systems.

**Figure 4 sensors-22-06451-f004:**
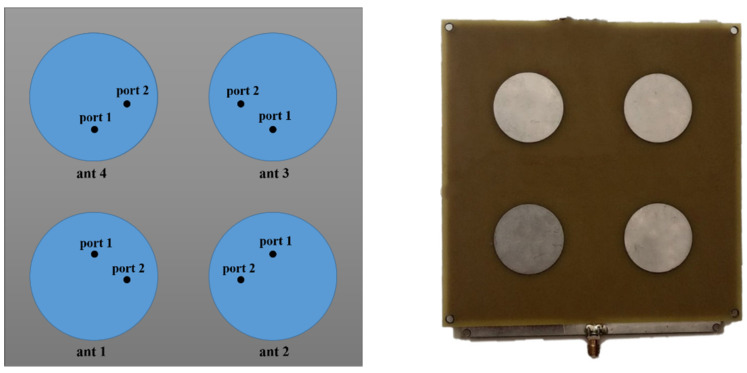
Topological diagram of the PAA with 2 × 2 patch elements and feed points.

**Figure 5 sensors-22-06451-f005:**
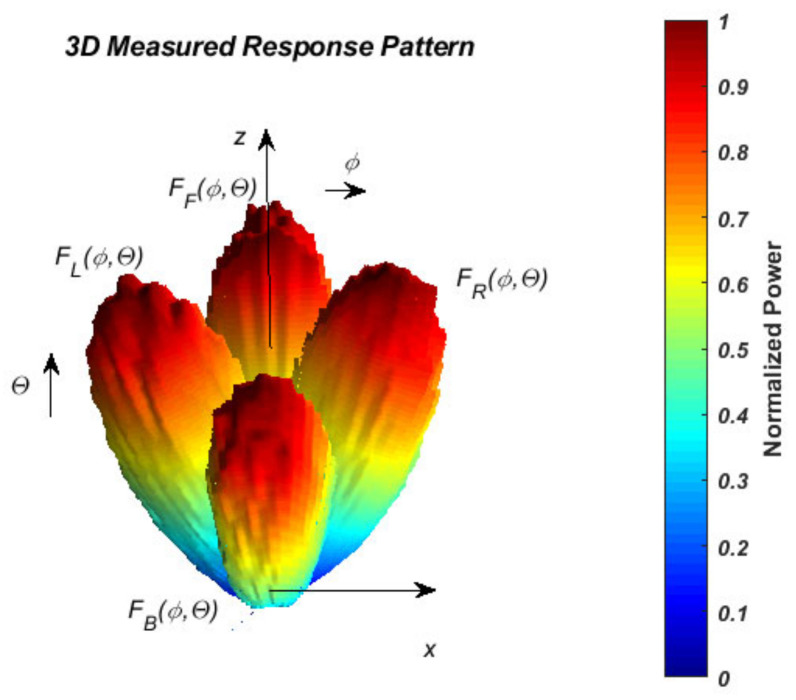
3D measured RP states 2–5.

**Figure 6 sensors-22-06451-f006:**
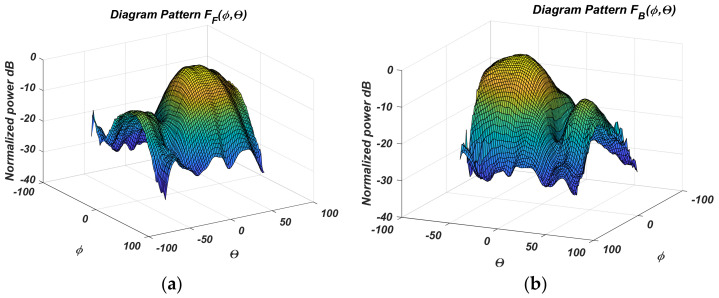
Measured relative antenna gain in Cartesian angular coordinates for: (**a**) st; (**b**) st.

**Figure 7 sensors-22-06451-f007:**
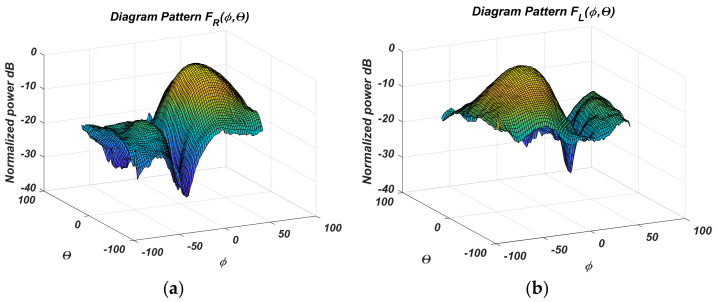
Measured relative antenna gain in Cartesian angular coordinates for: (**a**) st. 2; (**b**) st. 4.

**Figure 8 sensors-22-06451-f008:**
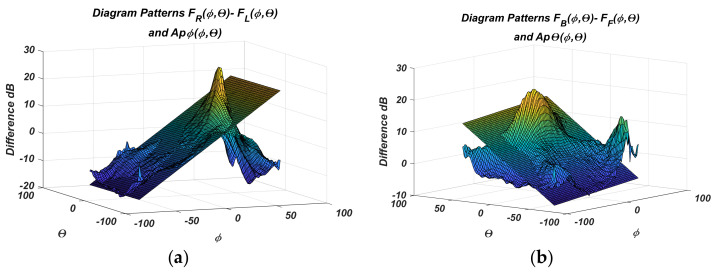
Difference diagrams in Cartesian angular coordinates for: (**a**) *F_R_*-*F_L_*; (**b**) *F_F_*-*F_B_*.

**Figure 9 sensors-22-06451-f009:**
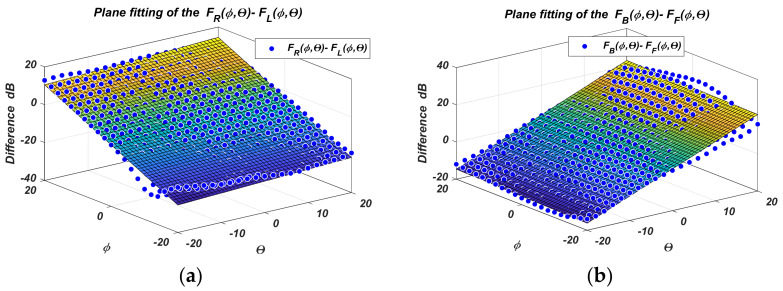
Linear regression of difference diagrams for: (**a**) *F_R_*-*F_L_*; (**b**) *F_F_*-*F_B_*.

**Figure 10 sensors-22-06451-f010:**
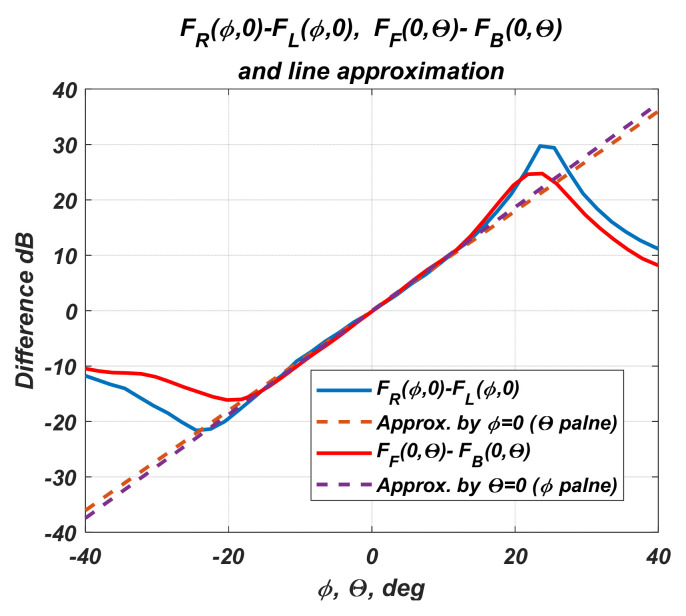
Difference diagrams in *ϕ* and θ planes and their approximations.

**Figure 11 sensors-22-06451-f011:**
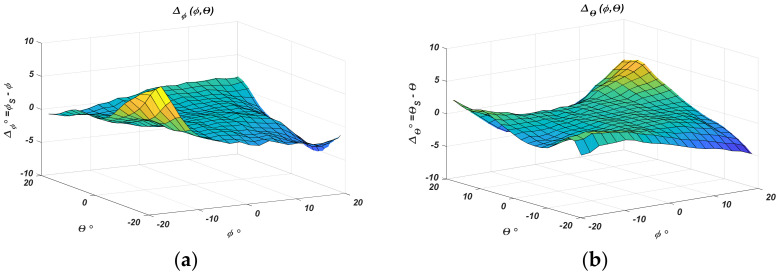
The absolute error: (**a**) *ϕ*; (**b**) *θ*.

**Figure 12 sensors-22-06451-f012:**
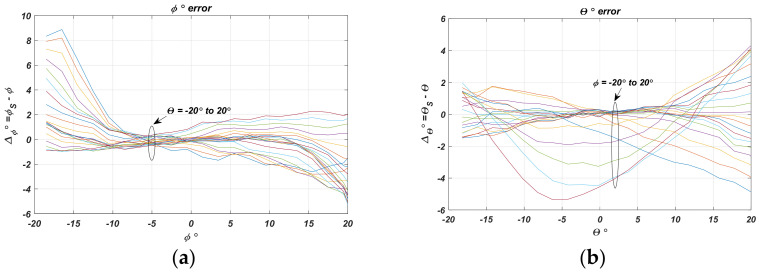
The absolute error: (**a**) *ϕ* at parameter *θ*; (**b**) *θ* at parameter *ϕ*.

**Figure 13 sensors-22-06451-f013:**
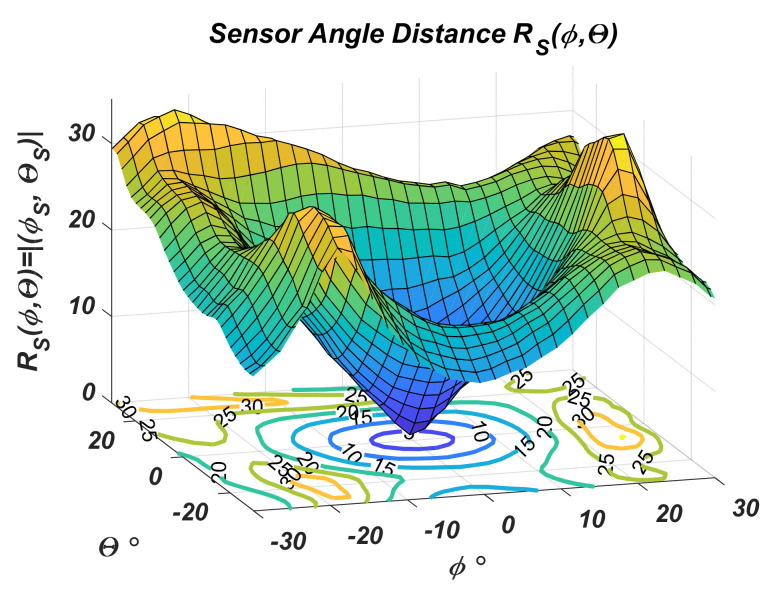
Evaluation of angular distance *R_S_* as a function of actual angular coordinates.

**Figure 14 sensors-22-06451-f014:**
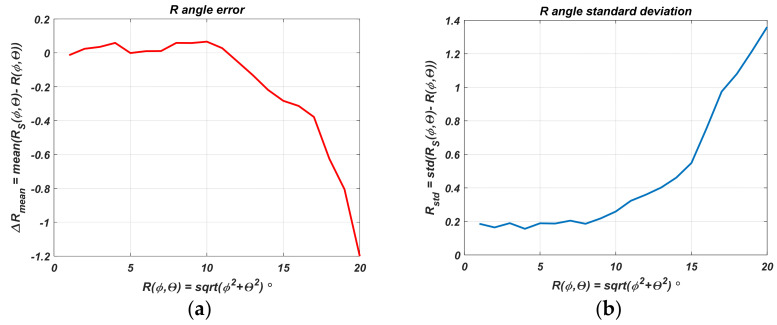
Average error of angular distance (**a**) and its standard deviation (**b**).

**Figure 15 sensors-22-06451-f015:**
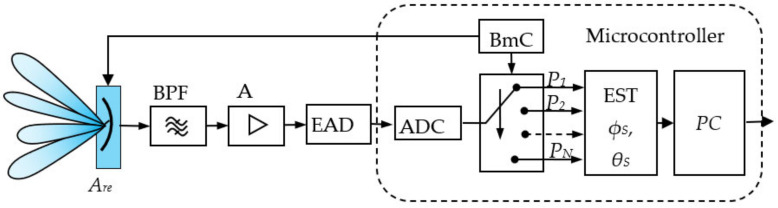
Studied radio receiving part of the PCS system.

**Figure 16 sensors-22-06451-f016:**
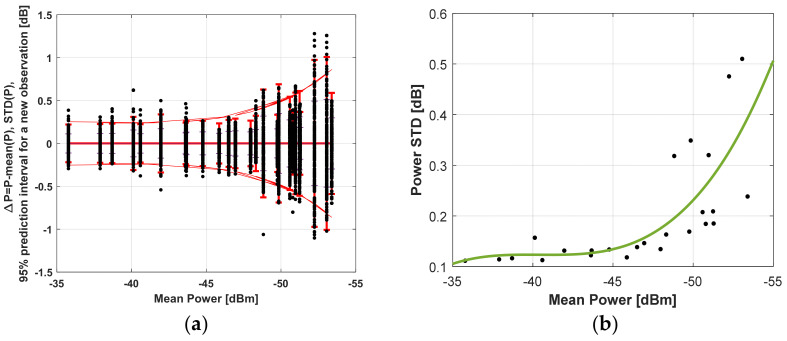
(**a**) Absolute deviation, standard deviation of the measured power and 95% prediction interval for the new measurement. (**b**) Standard deviation of the measured power and fitting curve v.s. mean power.

**Figure 17 sensors-22-06451-f017:**
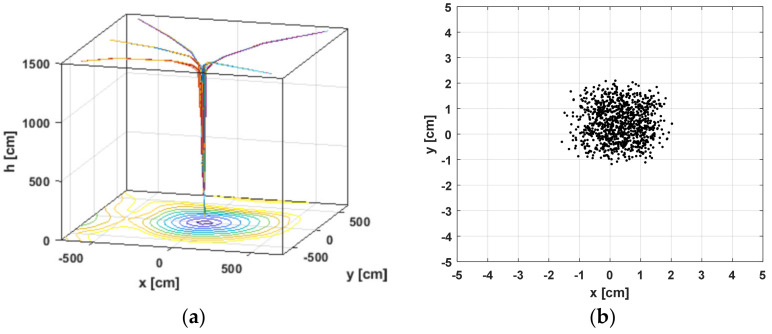
Landing algorithm simulation with initial UAV height *ho* = 15 m and object deviation of ±25°. (**a**) 3D landing trajectories; (**b**) Object coordinates for height *hmin* = 1 m above the sensor.

**Figure 18 sensors-22-06451-f018:**
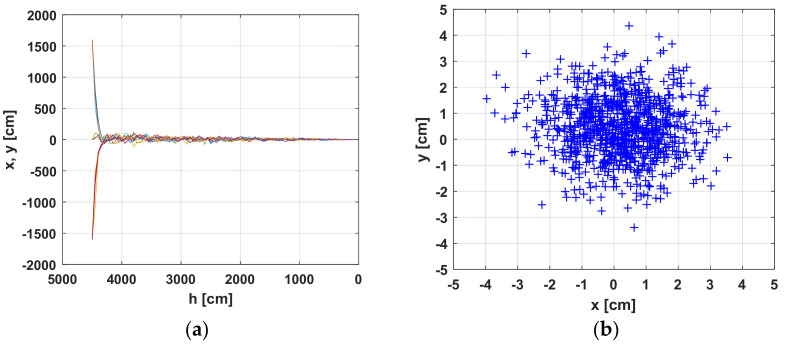
Simulation of the landing algorithm with initial UAV height *ho* = 45 m and initial object deviation of ±20. (**a**) 2D landing trajectories; (**b**) Object coordinates for height *hmin* = 1 m above the sensor.

**Table 2 sensors-22-06451-t002:** Absolute error from approximation.

*ϕ*, *θ* [deg]	Δ*ϕ*, Δ*θ* [deg]	Δ*x*, Δ*y* [m], *h* = 1 m
0°, 0°	0.11°, 0.12°	0.0019, 0.0021
−20 to 20°, −20 to 20°	max 8°, max 5.8°	0.14, 0.10
−15 to 15°, −15 to 15°	max 2°, max 2°	0.035, 0.035

## Data Availability

Not applicable.

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
