# Peer review of "Automatic Landing of Unmanned Aerial Vehicles via Wireless Positioning System with Pseudo-Conical Scanning"

_sensors, 2022, doi:10.3390/s22176451_

Round 1
Reviewer 1 Report
The paper discusses a radio-based landing algorithm for UAVs, with some interesting performance figures based on simulation and statistical results. While I am not an expert in navigation systems, the results seem solid and well-justified. The only thing I would suggest is a comparison with some of the state of the art solutions mentioned in the first sections, which would help identifying the performance gain. Also, there are several spelling and grammar mistakes throughout the paper and even in the abstract, the authors should perform a thorough proofreading. In some places, the writing makes the discussion difficult to follow, limiting the value of the paper.
Author Response
Thanks for the detailed review. We accept all the comments that have been fixed in the edited version.

Reviewer 2 Report
As indicated in the attached file.

Author Response

(The authors gave the same response as above.)
